# Quantum Treatment of Inelastic Interactions for the Modeling of Nanowire Field-Effect Transistors

**DOI:** 10.3390/ma13010060

**Published:** 2019-12-21

**Authors:** Youseung Lee, Demetrio Logoteta, Nicolas Cavassilas, Michel Lannoo, Mathieu Luisier, Marc Bescond

**Affiliations:** 1Integrated Systems Laboratory, ETH Zürich, 8092 Zürich, Switzerland; youseung.lee@iis.ee.ethz.ch (Y.L.); mluisier@iis.ee.ethz.ch (M.L.); 2IM2NP, UMR CNRS 7334, Aix-Marseille Université, Technopôle de Château-Gombert, Bâtiment Néel, 60 Rue Frédéric Joliot Curie, 13453 Marseille, France; logotetad@gmail.com (D.L.); nicolas.cavassilas@im2np.fr (N.C.); michel.lannoo@free.fr (M.L.); 3LIMMS, CNRS-UMI 2820, Institute of Industrial Science, University of Tokyo, Tokyo 153-8505, Japan

**Keywords:** quantum modeling, nonequilibrium Green’s function, nanowire transistor, electron–phonon interaction, phonon–phonon interaction, self-consistent Born approximation, lowest order approximation, Padé approximants, Richardson extrapolation

## Abstract

During the last decades, the Nonequilibrium Green’s function (NEGF) formalism has been proposed to develop nano-scaled device-simulation tools since it is especially convenient to deal with open device systems on a quantum-mechanical base and allows the treatment of inelastic scattering. In particular, it is able to account for inelastic effects on the electronic and thermal current, originating from the interactions of electron–phonon and phonon–phonon, respectively. However, the treatment of inelastic mechanisms within the NEGF framework usually relies on a numerically expensive scheme, implementing the self-consistent Born approximation (SCBA). In this article, we review an alternative approach, the so-called Lowest Order Approximation (LOA), which is realized by a rescaling technique and coupled with Padé approximants, to efficiently model inelastic scattering in nanostructures. Its main advantage is to provide a numerically efficient and physically meaningful quantum treatment of scattering processes. This approach is successfully applied to the three-dimensional (3D) atomistic quantum transport OMEN code to study the impact of electron–phonon and anharmonic phonon–phonon scattering in nanowire field-effect transistors. A reduction of the computational time by about ×6 for the electronic current and ×2 for the thermal current calculation is obtained. We also review the possibility to apply the first-order Richardson extrapolation to the Padé *N*/*N* − 1 sequence in order to accelerate the convergence of divergent LOA series. More in general, the reviewed approach shows the potentiality to significantly and systematically lighten the computational burden associated to the atomistic quantum simulations of dissipative transport in realistic 3D systems.

## 1. Introduction

Recent advances in the nanostructure engineering have led to a vast variety of nano-scale material applications in different areas, e.g., electronics [1], photonics [2], and thermoelectric devices [3]. Among them, nanowire (NW) field-effect transistor (FET) represents the most promising architecture for the next generation logic switches capable to reach sub-10-nm gate lengths in mass production [4]. The versatility of this architecture resides mostly in the fact that it can provide an excellent electrostatic control thanks to the gate-all-around (GAA) configuration [5] and can incorporate novel materials exhibiting high mobilities and/or high band gap [6,7,8,9,10,11].

In order to move forward in the development of NW FETs, various quantum mechanical effects inside NWs should be theoretically investigated. In line with that purpose, there have been abundant theoretical achievements and discussions to shed light on quantum physics underlying electrical transport. Wigner function-based quantum transport approach [12,13,14], considered as a quantum counterpart of Boltzmann transport equation, is one of such achievements. Pauli master equation was also proposed to treat inelastic scattering processes by using the concept of scattering states based on Fermi’s Golden rule [15,16]. Meanwhile, Bohmian mechanics has been applied to develop a many-body quantum transport simulator [17,18].

Along with all the methods aforementioned, the Nonequilibrium Green’s function (NEGF) formalism, first developed by Gordon Baym, Leo P. Kadanoff [19,20], and Leonid V. Keldysh [21] in the 1960s, has attracted intensive interests due to its capability to address various quantum effects taking place in nanostructures [22,23,24]. In particular, during the last decade there have been significant attempts to apply NEGF formalism for describing quantum mechanical effects inside nano-scaled devices such as quantum confinement [25], tunneling [26,27], surface roughness scattering [28], and electron–phonon interactions [29]. Among those effects, the treatment of inelastic interactions within this formalism, based on the concept of scattering self-energy, has become widespread.

From a numerical point of view, however, a challenge still remains in treating self-energies, which is particularly evident in applying NEGF to three-dimensional (3D) realistic structures. The main bottleneck usually comes from the conventional treatment of the nonlinear Dyson’s equation, within the self-consistent Born approximation (SCBA). Indeed, the SCBA satisfies the current conservation law via a Φ-derivable self-energy, that is, Σ[G]=δΦ/δG, but it requires a huge number of iterations to meet this condition. This is due, in part, to the innate characteristics of the SCBA algorithm, that works by including in the approximate solution higher-order Feynman diagrams at each iteration, regardless of if they are or not current conserving. The SCBA iteration process stops when the conservation law is satisfied, that is, when conserving diagrams are dominant over non-conserving ones. As a consequence, the implementation of the SCBA algorithm in atomistic 3D NEGF codes is usually only manageable with the help of supercomputer environments comprising several hundreds of CPUs/GPUs.

In this article, we review a highly efficient method [30,31,32,33,34], the so-called lowest order approximation (LOA) analytically continued by Padé approximants, to treat inelastic interactions within the NEGF framework. The main idea behind the method is to collect only the low-order scattering diagrams that guarantee the conservation of the current. Several analytic continuation techniques (e.g., Shanks transformation [35], Borel-Padé resummation [36], and hyper geometric resummation technique [37]) can be applied to the LOA series in order to reconstruct physical observables when the series is divergent. Here, we only focus on a rescaling technique [33,34] in combination with the Padé approximants [36,38], since it is very simple to implement in conventional NEGF codes. In the case of strongly divergent LOA series, we also show the application of the first-order Richardon extrapolation to the Padé N/N−1 approximants. The main advantage of this approach is that it avoids a large number of iterations while obtaining a relatively high degree of accuracy with respect to SCBA results.

To benchmark our method against the conventional SCBA scheme, we show its efficiency and accuracy in the calculation of electronic (including electron–phonon scattering) and phonon thermal (including anharmonic phonon–phonon scattering) currents flowing through a GAA NW-FET. Our investigation focuses on an n-type square cross-sectional silicon (Si) GAA NW-FET crystallographically oriented in 〈100〉 transport direction. As key findings, we show that the third-order LOA electronic currents analytically continued by Padé approximants can reproduce SCBA results within an error ⩽10%. A similar accuracy is found by applying the first-order Richardson extrapolation to the sequence of Padé approximants of the thermal current.

The rest of the paper is organized as follows: Section 2 describes the theory of the LOA analytically continued by Padé approximants and compares it to the conventional SCBA iterative scheme. A simple rescaling technique to calculate the LOA series directly from the SCBA algorithm will also be explained. In Section 3, we will show that the approach can be advantageously applied to the computation of the electron and phonon currents of GAA NW-FETs. Finally, Section 4 will conclude the article with the key findings and the outlook of the method.

## 2. General Theoretical Framework

In this section, we theoretically detail the LOA approach for the treatment of inelastic interactions within the NEGF framework [29,39,40,41] and we compare it to the conventional SCBA. We then present a simple rescaling technique, computationally most efficient and applicable to 3D realistic structures, to calculate the LOA expectation values from the conventional SCBA method. A matrix form of Padé approximants and the first-order Richardson extrapolation technique are also discussed.

### 2.1. Dyson Equation

In the NEGF theory, the Dyson equation links the noninteracting Green’s function $g_{0}$ to the fully interacting Green’s function *G*
via the scattering self-energy Σ, which describes all the interactions. In a simplified matrix notation, we have
(1)$$G=g_{0}+g_{0}\Sigma \left[G\right]G,$$
with the abbreviations i = ($r_{i}$, $t_{i}$), $g_{0}\Sigma \left[G\right]G=\int \;d2\int \;d2^{\prime }\;g_{0}\left(1;2\right)\Sigma \left(2;2^{\prime }\right)G\left(2^{\prime };1^{\prime }\right)$. The Feynman diagrams corresponding to electron–phonon scattering are shown in Figure 1. Since the scattering self-energy is a functional of *G*, i.e., Σ=Σ[G], the Dyson equation is nonlinear and suitable approximations are required for the self-energy. The Φ-derivable approximation in the Luttinger–Ward picture [42,43] is a prescription to construct a self-energy satisfying the current conservation law. Proper resolutions of the Dyson equation should then be handled with a Φ-derivable self-energy. While both the conventional SCBA and the LOA provide a Φ-derivable self-energy, they solve the Dyson equation by using two different kind of algorithms, iterative and direct, respectively.

### 2.2. Self-Consistent Born Approximation

The nonlinearity of the Dyson equation leads itself naturally to an iterative scheme to obtain solutions. This inspiration then led to the application of the Born approximation in the concept of self-consistency, i.e., SCBA, which is commonly considered essential for Φ-derivability. By assuming $G_{N}≃G_{N-1}$ with very large *N*, Equation (Equation 1) can be rewritten as
(2)$$G_{N}={\left[g_{0}^{-1}-\Sigma \left[G_{N-1}\right]\right]}^{-1},$$
or in a Taylor series expansion form
(3)$$\begin{array}{c} \begin{array}{cc} G_{N}= & g_{0}+g_{0}\Sigma \left[G_{N-1}\right]g_{0}+g_{0}\Sigma \left[G_{N-1}\right]g_{0}\Sigma \left[G_{N-1}\right]g_{0}+\cdots . \end{array} \end{array}$$
where $G_{N}$ stands for the Green’s function at the *N*th iteration step. The SCBA Green’s functions $G_{1}$ and $G_{2}$ are then defined from Equation (Equation 3) as
(4)$$G_{1}=g_{0}+g_{0}\Sigma \left[g_{0}\right]g_{0}+g_{0}\Sigma \left[g_{0}\right]g_{0}\Sigma \left[g_{0}\right]g_{0}+\cdots .$$
and
(5)$$\begin{array}{c} \begin{array}{cc} G_{2}= & g_{0}+g_{0}\Sigma \left[G_{1}\right]g_{0}+g_{0}\Sigma \left[G_{1}\right]g_{0}\Sigma \left[G_{1}\right]g_{0}+\cdots ,\\ = & g_{0}+g_{0}\Sigma \left[g_{0}\right]g_{0}+\left(g_{0}\Sigma \left[g_{0}\right]g_{0}\Sigma \left[g_{0}\right]g_{0}+g_{0}\Sigma \left[g_{0}\Sigma \left[g_{0}\right]g_{0}\right]g_{0}\right)\\ + & g_{0}\Sigma \left[g_{0}\right]g_{0}\Sigma \left[g_{0}\right]g_{0}\Sigma \left[g_{0}\right]g_{0}+\cdots , \end{array} \end{array}$$
respectively. As shown in Figure 2a,b, each SCBA Green’s function includes an infinite number of diagrams. Therefore, $G_{1}$ and $G_{2}$ do not necessarily preserve the conservation law since there are non-conserving higher-order diagrams, according to the corresponding scattering order [31]. This character of the SCBA scheme allows only an asymptotic approach to the conservation law, that is, a large number of iterations with specified convergence criteria depending on the scattering strength of the system.

### 2.3. Lowest Order Approximation

The fact that the SCBA Green’s functions include non-conserving diagrams has inspired the concept of the LOA method, i.e., calculating only the conserving diagrams based on the Φ-derivability at each scattering order, as shown in Figure 3a,b. The first-order LOA in the case of electron–phonon scattering has been suggested in Reference [30] by treating the term $g_{0}\Sigma \left[G\right]G$ from Equation (Equation 1) as a perturbation δG and by applying a Taylor series expansion to the corresponding self-energy to obtain the first-order LOA Green’s function $g_{1LOA}$ as
(6)$$g_{1LOA}=g_{0}+g_{0}\Sigma \left[g_{0}\right]g_{0},$$
where the interaction self-energy is constructed only by the noninteracting Green’s function $g_{0}$ and is Φ-derivable since $\Sigma \left[g_{0}\right]=\delta \Phi \left[g_{0}\right]/\delta g_{0}$. The work in Reference [30] showed that, in the weak-scattering regime, the expectation value constructed from the first-order LOA Green’s function (therein, 1st LOA current) is very similar to the one from the SCBA. However, it was also shown that the spectral currents built by the 1st LOA Green’s function are very far from the SCBA results since only the first order in the interaction was included.

In References [31,32], it was shown that, by using the term $\Delta g_{1}=g_{0}\Sigma \left[g_{0}\right]g_{0}$ from Equation (Equation 6) as a basic building block, the generalized LOA expansion series can be built as
(7)$$g_{N}=g_{0}+\sum_{n=1}^{N}\Delta g_{n},$$
where $g_{N}\left(N>1\right)$ is the LOA Green’s function at *N*th-order in the interaction and $\Delta g_{n}$ is the perturbation term of order *n* in the interaction. Since the Dyson equation has a recursion relation $g_{0}\Sigma \left[G\right]G$, the general expression of $g_{N}$ can be obtained by injecting Equation (Equation 7) into Equation (Equation 1) such that
(8)$$g_{0}+g_{0}\Sigma \left[g_{N-1}\right]g_{N-1}=g_{0}+g_{0}\Sigma \left[g_{0}+\sum_{n=1}^{N-1}\Delta g_{n}\right]\;\left(g_{0}+\sum_{n=1}^{N-1}\Delta g_{n}\right),$$
and then by discarding all higher-order terms to include only *N*-order interactions for $g_{N}$. For example, by following the previous process, $g_{2}$ and $g_{3}$ are constructed as
(9)$$g_{2}=g_{1}+g_{0}\Sigma_{1}\Delta g_{1}+g_{0}\Sigma_{2}\Delta g_{0}=g_{1}+\Delta g_{2},$$
(10)$$g_{3}=g_{2}+g_{0}\Sigma_{1}\Delta g_{2}+g_{0}\Sigma_{2}\Delta g_{1}+g_{0}\Sigma_{3}\Delta g_{0}=g_{2}+\Delta g_{3},$$
where the corresponding self-energy is defined such that $\Sigma_{n}$ depends on the n−1 order perturbation term
(11)$$\Sigma_{n}=\Sigma \left[\Delta g_{n-1}\right].$$

As a result, the perturbation term for N⩾1 is as follows:(12)$$\Delta g_{N}=g_{0}\sum_{n=0}^{N-1}\Sigma_{N-n}\Delta g_{n},$$
with
(13)$$\Delta g_{1}=g_{0}\Sigma \left[g_{0}\right]g_{0},$$
and
(14)$$g_{N}=g_{N-1}+\Delta g_{N}.$$

The overall schematic of practical calculations for the LOA Green’s function $g_{N}$, which was explained in Reference [32] in detail, is summarized in Algorithm 1 below.
**Algorithm 1***N*th-order LOA calculation. N← <the order of LOA> **for**
i←0toN
**do**  **if**
i=0
**then**    $g_{0}^{r}\leftarrow \left[EI-H-\Sigma_{C}^{r}\right]$, *E* (Energy), *H* (Hamiltonian), and $\Sigma_{C}$ (Contact self-energy)    $g_{0}^{≶}\leftarrow g_{0}^{r}\Sigma_{C}^{≶}g_{0}^{a}$, the superscript *r* (Retarded), *a* (Advanced), and ≶ (Lesser/Greater)    $\Delta g_{0}^{≶}\leftarrow g_{0}^{≶},\Delta g_{0}^{r}\leftarrow g_{0}^{r}$  **else**    **for**
n←0toi−1
**do**      : interacting self energy calculation      $\Sigma_{int,\;i}^{≶}$ and $\Sigma_{int,\;i}^{r}$      : perturbation term calculation      $temp\left(\Delta g_{i}^{r}\right)\leftarrow g_{0}^{r}\Sigma_{int,\;i-n}^{r}\Delta g_{n}^{r}$      ⇓ applying Langreth Theorem      $temp\left(\Delta g_{i}^{≶}\right)\leftarrow g_{0}^{r}\Sigma_{int,\;i-n}^{r}\Delta g_{n}^{≶}+g_{0}^{r}\Sigma_{int,\;i-n}^{≶}\Delta g_{n}^{a}+g_{0}^{≶}\Sigma_{int,\;i-n}^{a}\Delta g_{n}^{a}$      $\Delta g_{i}^{r}\leftarrow \Delta g_{i}^{r}+temp\left(\Delta g_{i}^{r}\right)$      $\Delta g_{i}^{≶}\leftarrow \Delta g_{i}^{≶}+temp\left(\Delta g_{i}^{≶}\right)$    **end for**    $g_{i}^{r}\leftarrow g_{i-1}^{r}+\Delta g_{i}^{r}$    $g_{i}^{≶}\leftarrow g_{i-1}^{≶}+\Delta g_{i}^{≶}$  **end if** **end for**

Equations (Equation 12)–(Equation 14) show that, by construction, each LOA Green’s function $g_{N}$ includes only *N*-order interactions, meaning that it can generate the corresponding expectation value $O_{N}=O\left(g_{N}\right)$. We then obtain the LOA series for any expectation value. For example, in this article, electronic current series (including electron–phonon scattering) and thermal current series (including phonon–phonon scattering) are defined as $I_{N}=I\left(g_{N}\right)$ and $Q_{N}=Q\left(g_{N}\right)$, respectively, to any order in the interaction:(15)$$I_{N}=I_{0}+\sum_{n=1}^{N}\Delta I_{n},\;\;Q_{N}=Q_{0}+\sum_{n=1}^{N}\Delta Q_{n},$$
where $\Delta I_{n}=I_{n}-I_{n-1}$ and $\Delta Q_{n}=Q_{n}-Q_{n-1}$ represent the difference between the expectation values of the *n*th and $\left(n-1\right)$th orders and where $I_{0}$ and $Q_{0}$ are the expectation values from the noninteracting Green’s function $g_{0}$ (here, the ballistic one).

Indeed, with Algorithm 1, one can compute the exact LOA Green’s function at any order within the considered interaction. However, the Langreth theorem, involved in transforming the contour integration into real-time integration, leads to a large number of matrix inversions and multiplications for higher-order LOA calculations, which restricts the applicability of the method. Therefore, in the next section, we present a simple rescaling technique to directly calculate the LOA expectation values from the SCBA results.

### 2.4. Rescaling Technique

The rescaling technique has been developed in Reference [33] to overcome the limitation of Algorithm 1 in the spirit of preserving the conventional SCBA algorithm since the latter algorithm is computationally cheap as long as a small number of iterations is needed. The method was applied to the case of electron–phonon scattering inside GAA NW-FETs, described by using an $sp^{3}d^{5}s^{*}$ tight-binding (TB) model, as implemented in the atomistic code OMEN [29,44]. Later on, Lee et al. [34] have shown that the method is also applicable to describe anharmonic phonon–phonon scattering in which the anharmonic self-energy Π is a function of the square of phonon Green’s functions *D*.

The basic idea of the rescaling technique is that, by multiplying a properly chosen factor 1/λ to any scattering self-energy (e.g., Σ[G]/λ for electron–phonon and Π[DD]/λ for phonon–phonon), infinite non-conserving diagrams in SCBA Green’s functions can be eliminated. The reason behind this is based on the fact that conserving and non-conserving diagrams in the SCBA are arranged according to ascending order in interactions, as shown in Figure 2.

For example, let us rewrite Equation (Equation 3), which is a Taylor expansion series of the SCBA Green’s function, in a general form applicable to any Green’s function (Z=G for electron and Z=D for phonon), as
(16)$$\begin{array}{c} \begin{array}{cc} Z_{N}= & z_{0}+z_{0}\Phi \left[{\left(Z_{N-1}\right)}^{l}\right]z_{0}\\ & +z_{0}\Phi \left[{\left(Z_{N-1}\right)}^{l}\right]z_{0}\Phi \left[{\left(Z_{N-1}\right)}^{l}\right]z_{0}+\cdots ,\;\;l⩾1, \end{array} \end{array}$$
where $z_{0}$ is the noninteracting Green’s function and *l* denotes the dependence of the self-energy on the Green’s function to the *l*th power ($\Phi \left[Z_{N-1}\right]=\Sigma \left[G_{N-1}\right]$ with l=1 for electron–phonon and $\Phi \left[Z_{N-1}Z_{N-1}\right]=\Pi \left[D_{N-1}D_{N-1}\right]$ with l=2 for phonon–phonon). By introducing a scaling parameter $\lambda_{1}$ into the scattering self-energy and by proceeding to the first SCBA iteration, higher-order non-conserving diagrams vanish [33] since each term is scaled by ${\left(1/\lambda_{1}\right)}^{n}$ in the interaction order *n*. We then express the first-order LOA Green’s function rescaled by $\lambda_{1}$ as
(17)$$Z_{1}^{\lambda_{1}}=z_{0}+\frac{1}{\lambda_{1}}\Delta z_{1}\left(=z_{0}\Phi_{1}z_{0}\right),\;\;\Phi_{1}=\Phi \left[{\left(z_{0}\right)}^{l}\right],$$
where the *n*th-order (n>1) non-conserving terms are suppressed thanks to the factors ${\left(1/\lambda_{1}\right)}^{n}$. The first-order LOA expectation value ($O_{1}$) is then calculated by using $\lambda_{1}$ such as $O_{1}=O_{0}+\Delta O_{1}=O_{0}\left(z_{0}\right)+O\left(z_{0}\Phi \left[{\left(z_{0}\right)}^{l}\right]z_{0}\right)=O_{0}\left(z_{0}\right)+\lambda_{1}\left[O\left(Z_{1}^{\lambda_{1}}\right)-O\left(z_{0}\right)\right]$.

For the second-order calculation, let us assume that we have the first-order LOA Green’s function rescaled by $\lambda_{2}$ as $Z_{1}^{\lambda_{2}}=z_{0}+\frac{1}{\lambda_{2}}\Delta z_{1}$. The second SCBA iteration with a properly chosen scaling factor $\lambda_{2}$ then produces the self-energy $\Phi_{2}$ as
(18)$$\begin{array}{c} \begin{array}{cc} \Phi_{2}= & \Phi \left[\frac{{\left(Z_{1}^{\lambda_{2}}\right)}^{l}}{\lambda_{2}}\right]=\Phi \left[\frac{1}{\lambda_{2}}{\left(z_{0}+\frac{1}{\lambda_{2}}\Delta z_{1}\right)}^{l}\right],\\ = & \Phi \left[\frac{{\left(z_{0}\right)}^{l}}{\lambda_{2}}+\frac{z_{0}^{l-1}\Delta z_{1}}{\lambda_{2}^{2}}+\cdots +\frac{z_{0}\Delta z_{1}^{l-1}}{\lambda_{2}^{l}}+\frac{{\left(\Delta z_{1}\right)}^{l}}{\lambda_{2}^{l+1}}\right]. \end{array} \end{array}$$

In Equation (Equation 18), only the terms related to $\left(1/\lambda_{2}\right)$ and ${\left(1/\lambda_{2}\right)}^{2}$ are retained while the other terms are omitted. Therefore, the second-order LOA Green’s function $Z_{2}^{\lambda_{2}}$ rescaled by $\lambda_{2}$ also does not include all the terms related to ${\left(1/\lambda_{2}\right)}^{N}$ (N>2). Reconstructing the second-order LOA expectation value then follows the way explained in Reference [33]. This process can be generalized to any order LOA Green’s function. Given an exact Green’s function at the (N−1)th order with the rescaling factor $\lambda_{N}$ as
(19)$$Z_{N-1}^{\lambda_{N}}=z_{0}+\frac{1}{\lambda_{N}}\Delta z_{1}+\cdots +\frac{1}{\lambda_{N}^{N-1}}\Delta z_{N-1},$$
an exact self-energy at the *N*th-order can be obtained since the scaling factor $\lambda_{N}$ automatically cancels out all the higher-order interactions (>Nth-order): if $Z_{N-1}$ is correct to the (N−1)th order, $\Phi_{N}\left[{\left(Z_{N-1}\right)}^{l}\right]$ is also correct to the *N*th-order [34]. By using the proposed method, any-order LOA expectation values can then be easily obtained from the SCBA iterations.

### 2.5. Matrix Form of the Padé Approximants

The LOA Green’s function, computed either by the direct algorithm or by the rescaling technique, consists of finite conserving diagrams up to the *N*th-order, i.e., forms a truncated perturbation-expansion series. This series depends on an interaction parameter *U*, always associated to two vertices in the electron–phonon and phonon–phonon scattering diagrams:(20)$$z_{0}+\Delta z_{1}\left(U\right)+\cdots +\Delta z_{N}\left(U^{N}\right).$$

As in the general case of perturbation series, the LOA series has a radius of convergence $U_{r}$, then being convergent ($U<U_{r}$) or divergent ($U>U_{r}$), depending on *U* [36,38,45]. Therefore, in order to obtain meaningful values for the physical observables, the application of resummation techniques (e.g., Padé approximants [31,32,33,34,46,47], hypergeometric resummation [37], and Padé + Richardson extrapolation [34]) is generally required. Here, we review a matrix form of Padé approximants [48] since it has been already proved to be very efficient with a relatively high accuracy in the previous works [32,33,34].

The Padé approximant transforms a power series function $F_{N}\left(x\right)$ into a fractional function including two polynomials, $P_{p}\left(x\right)$ as the numerator and $M_{m}\left(x\right)$ as the denominator [49] such as
(21)$$F_{N}\left(x\right)=\sum_{k=0}^{p+m}f_{k}x^{k}=f_{0}+f_{1}x+f_{2}x^{2}+\cdots +f_{p+m}x^{p+m}=\frac{p_{0}+p_{1}x+p_{2}x^{2}+\cdots +p_{p}x^{p}}{1+m_{1}x+m_{2}x^{2}+\cdots +m_{m}x^{m}}=F_{p/m}\left(x\right).$$

By matching coefficients at the same order between two functions, l+m+1 linear homogeneous equations can be obtained as
(22)$$\begin{array}{cc} p_{0} & =f_{0},\\ p_{1}-f_{0}m_{1} & =f_{1},\\ p_{2}-f_{0}m_{2}-f_{1}m_{1} & =f_{2},\\ \vdots \\ p_{p}-f_{0}m_{p}-f_{1}m_{p-1}-\cdots -f_{p-1}m_{1} & =f_{p},\\ -f_{p-m+1}m_{m}-f_{p-m+2}m_{m-1}-\cdots -f_{p}m_{1} & =f_{p+1},\\ -f_{p-m+2}m_{m}-f_{p-m+3}m_{m-1}-\cdots -f_{p+1}m_{1} & =f_{p+2},\\ \vdots \\ -f_{p}m_{m}-f_{p+1}m_{m-1}-\cdots -f_{p+m-1}m_{1} & =f_{p+m}. \end{array}$$

We then obtain a matrix form Ax=b to calculate Padé coefficients $p_{0},\ldots ,p_{p}$ and $m_{1},\ldots ,m_{m}$ as
(23)$$\left[\begin{array}{cccccccc} 1 & 0 & \cdots & 0 & 0 & 0 & \cdots & 0\\ 0 & 1 & \cdots & 0 & -f_{0} & 0 & \cdots & 0\\ \vdots & \; & \ddots & \vdots & \vdots & \; & \ddots & \vdots \\ 0 & 0 & \cdots & 1 & -f_{p-1} & -f_{p-2} & \cdots & 0\\ 0 & 0 & \cdots & 0 & -f_{p} & -f_{p-1} & \cdots & -f_{p-m+1}\\ 0 & 0 & \cdots & 0 & -f_{p+1} & -f_{p} & \cdots & -f_{p-m+2}\\ \vdots & \; & \ddots & \vdots & \vdots & \; & \ddots & \vdots \\ 0 & 0 & \cdots & 0 & -f_{p+m-1} & -f_{p+m-2} & \cdots & -f_{p} \end{array}\right]\left[\begin{array}{c} p_{0}\\ p_{1}\\ \vdots \\ p_{p}\\ m_{1}\\ m_{2}\\ \vdots \\ m_{m} \end{array}\right]=\left[\begin{array}{c} f_{0}\\ f_{1}\\ \vdots \\ f_{p}\\ f_{p+1}\\ f_{p+2}\\ \vdots \\ f_{p+m} \end{array}\right].$$

By solving Equation (Equation 23), a power series-transformed fractional function can be determined as $F_{p/m}\left(x\right)=\frac{p_{0}+p_{1}x+p_{2}x^{2}+\cdots +p_{p}x^{p}}{1+m_{1}x+m_{2}x^{2}+\cdots +m_{m}x^{m}}$. By appling this technique to the *N*th-order LOA expectation values $O_{N}$, Padé analytically continued expectation values $O_{p/m}$ with N=p+m are obtained.

### 2.6. Richardson Extrapolation

In this subsection, we introduce the first-order Richardson extrapolation technique, applicable to accelerate the convergence of divergent LOA expectation values analytically continuated by Padé N−1/N approximants toward SCBA values [34].

Let us assume that the LOA + Padé sequence is a monotone series $S_{N}$ approaching the SCBA result *S*. We then write $S_{N}$ in an asymptotic form as [50]
(24)$$S_{N}\approx S+\frac{c_{1}}{N}+\frac{c_{2}}{N^{2}}+\frac{c_{3}}{N^{3}}+\cdots ,$$
with unknown coefficients $c_{1}$, $c_{2}$, ⋯. By considering the first-order approximation of *S* based on two consecutive terms $S_{N}$ and $S_{N+1}$,
(25)$$N\left(S_{N}-S\right)=c_{1},\;\left(N+1\right)\left(S_{N+1}-S\right)=c_{1},$$
the first-order Richardson extrapolation can be obtained as
(26)$$S_{N}^{\left[1\right]}=\left(N+1\right)S_{N+1}-NS_{N}.$$

This technique can be applied to the sequence of LOA expectation values analytically continuated by the Padé N−1/N approximants ($O_{N-1/N}$). For example, the expectation values obtained from Padé 0/1($O_{0/1}$) and Padé 1/2 ($O_{1/2}$) become the first and second elements of the sequence, i.e., $S_{1}=O_{0/1}$ and $S_{2}=O_{1/2}$, respectively (in general, $S_{N}=O_{N-1/N}$). By using Equation (Equation 26) with $O_{0/1}$ and $O_{1/2}$, we can then predict SCBA values as
(27)$$S_{1}^{\left[1\right]}=2S_{2}-S_{1}=2O_{1/2}-O_{0/1}\approx O_{SCBA}.$$

## 3. Applications to Electron and Phonon Transports in a Nanowire Transistor

In this section, we benchmark the performance of the LOA + Padé approach against the SCBA scheme in the description of electron–phonon and anharmonic phonon–phonon scattering inside a 3D nano-device. The investigated device is a Si GAA NW-FET with a 3 nm × 3 nm square cross section where electron or phonon transport occurs along the 〈100〉 crystallographic direction (Figure 4). In such a small cross-sectional NW, the transport is dominated by quantum confinement effects. The corresponding electronic band-structure and phonon dispersion relation are shown in Figure 5a,b, respectively.

The coupling effects between electronic and anharmonic thermal currents are not considered since the aim of the benchmarking is only to test the efficiency and accuracy of the method for each scattering mechanism. The electronic and thermal currents are then separately calculated in different configurations as follows. (i) For electron transport, the gate length is $L_{G}=13$ nm while the source and drain extension lengths are $L_{S/D}=10$ nm, with a doping concentration of donors $N_{S/D}=1\times 10^{20}\;\;{\mathrm{cm}}^{-3}$. A 1-nm-thick silicon dioxide layer surrounds the NW structure. (ii) In the case of phonon transport, an ungated 60-nm-long NW structure without oxide layers is considered. All atoms on the NW surface are free to move. For the sake of clarity, only undoped NWs are considered. Under this condition, a very small number of electrons participate in the thermal conduction. This allows us to safely neglect the impact of eletron–phonon scattering [51].

In order to validate the LOA + Padé approach, we use the 3D atomistic NEGF code OMEN [29,44,51,52,53] in which a $sp^{3}d^{5}s^{*}$ TB model for electrons and a modified valence-force-field (VFF) method for phonons are implemented. OMEN is among the most sophisticated atomistic simulators of quantum transport of nano-devices. Indeed, the latest version of OMEN also includes density-functional theory (DFT)-based Hamiltonian expressed in a maximally localized Wannier function basis [10]. However, it requires considerable computational resources, especially when the SCBA loop is brought into play. In the following Section 3.1 and Section 3.2, the LOA approach within the rescaling technique formulation is used to calculate the electronic and thermal current. These results are benchmarked against the values obtained by running the full SCBA loop in OMEN until the current is conserved within a tolerance of 1%.

### 3.1. Electron–Phonon Scattering in a Nanowire Transistor

Here, we investigate steady-state electron transport in which electrons are coupled with the equilibrium phonon bath at room temperature. A full band *sp*${}^{3}$*d*${}^{5}$*s*${}^{*}$ TB model without spin-orbit coupling [54,55] is employed to describe the electronic states while a modified VFF method including four bond-interaction terms (harmonic approximations) [56,57] is used for the description of the NW phonon bath. The diagonal lesser self-energy for electron-phonon scattering is
(28)$$\Sigma_{nn}^{<}\left(E\right)=i\sum_{l}\sum_{\lambda ,q}\int_{-\infty }^{\infty }\frac{d(E^{\prime })}{2\pi }M_{nl}^{\lambda }\left(q\right)d_{0,\lambda }^{<}\left(q,E-E^{\prime }\right)G_{ll}^{<}\left(E^{\prime }\right)M_{ln}^{\lambda *}\left(q\right),$$
with
(29)$$M_{nl}^{\lambda }\left(q\right)=\sqrt{\frac{\hbar }{2\omega_{\lambda }\left(q\right)}}\sum_{i}\nabla_{i}H_{nl}\left[\frac{f_{\lambda }^{i}\left(R_{l},q\right)}{\sqrt{m_{l}}}-\frac{f_{\lambda }^{i}\left(R_{n},q\right)}{\sqrt{m_{n}}}\right],$$
and
(30)$$d_{0,\lambda }^{<}\left(q,\hbar \omega \right)=-2\pi i\left[\langle n_{\lambda }\left(q\right)\rangle \delta \left(\hbar \omega \mp \hbar \omega_{\lambda }\left(q\right)\right)+\left(\langle n_{\lambda }\left(q\right)\rangle +1\right)\delta \left(\hbar \omega \pm \hbar \omega_{\lambda }\left(q\right)\right)\right],$$
where *ℏ* is the reduced Planck constant, *E* is the electron energy, R is the position vector, *m* is the atomic mass, *f* is the phonon displacement, and ω is the phonon frequency. The indices *n* and *l* denote the atomic positions, while λ and q represent the phonon mode λ with momentum q. $\langle n_{\lambda }\left(q\right)\rangle$ is the expectation value of the phonon occupation in the mode λ and momentum q in thermal equilibrium. In the scattering self-energy Equation (Equation 28), the lesser Green’s function $G^{<}$ is coupled via the nearest-neighbor matrix elements M to the lesser phonon Green’s function $d_{0}^{<}$. The coupling matrix elements are obtained from the first derivative of the TB Hamiltonian matrix $\nabla_{i}H_{nl}$ between atoms *n* and *l* along the *i*th direction (*x*, *y*, or *z*). The real part of the self-energy, as commonly assumed in quantum transport nano-device modeling [58], is neglected.

By introducing three different scaling factors $\lambda_{1}$, $\lambda_{2}$, and $\lambda_{3}$ into Equation (Equation 28), as explained in Section 2.4, the LOA expectation values for electronic currents up to the third order can be obtained over a wide range of gate biases. The Padé approximant is then applied to analytically continue the LOA results. Figure 6 shows the drain current ($I_{D}$) vs. gate voltage ($V_{G}$) transfer characteristics obtained in the ballistic regime and by the SCBA, the 1st LOA, and the LOA + Padé 0/1 and 1/2 technique. It is clearly shown that 1st- and 3rd-order LOA currents combined with Padé 0/1 and 1/2, respectively, provide a fair evaluation of the SCBA result over the whole applied bias. The 1st LOA current is also very similar to the SCBA current except at high bias (i.e., $V_{G}=0.6$ V), due to the particularly strong electron–phonon scattering in this regime [29]. For this strong scattering regime, the Richardson extrapolation has been applied, thereby obtaining an accelerated convergence of the strongly divergent LOA series. More precisely, the accuracies at $V_{G}=0.6$
V are 89%, 93%, and 98% for the Padé 0/1, Padé 1/2, and Richardson, respectively. The values of the current at $V_{G}=0.6$
V obtained within the different approximations are reported in Table 1, along with the required computational burden in terms of number of SCBA iterations. These values show that the LOA approach combined with the Padé analytical continuation and the Richardson extrapolation is able to approximate the SCBA results within an error of 2% in 6 SCBA iterations, while the full SCBA loop needs 35 iterations to converge.

In a previous work, this approach has been applied to n- and p-type circular Si and Ge GAA NW FETs along the three main crystallographic orientations (<100>,<110>,<111>). A general similar conclusion about the efficiency of the LOA + Padé 1/2 has been reported, with a warning signal concerning the accuracy of the present approach for the situation where the 1st LOA current is larger than the ballistic one. Please refer to Reference [33] for the details.

### 3.2. Anharmonic Phonon–Phonon Scattering in A Nanowire

In this subseciton, we apply the LOA method to the steady-state anharmonic phonon transport. As described in Reference [51], a modified VFF method with harmonic bond interactions is used to construct the dynamical matrix, by which phonon frequencies are calculated. The noninteracting Green’s function (here, the ballistic Green’s function) is then calculated from the dynamical matrix and corresponding phonon frequency with the open boundary self-energy. Anharmonic effects, usually attributed to the third- and fourth-order contributions of the quantized atomic displacement [22,59] are described by a perturbative approach, that is, by solving the Dyson equation with the scattering self-energy in which VFF anharmonic interactions are included.

In this work, we focus on a single second-order perturbation effect based on the third-order anharmonicity, which models the anharmonic decay of a high-energy phonon ($\omega +\omega^{\prime }$) into two lower energy phonons (ω and $\omega^{\prime }$) or vice versa [51,59]. According to the contribution of the corresponding order, the greater/lesser anharmonic phonon–phonon scattering self-energy can be written as follows [51]:(31)$$\begin{array}{c} \begin{array}{cc} \Pi_{\nu \sigma }^{≷S,\;l_{2}^{\;}l_{3}^{\;}}\left(\omega \right)=2i\hbar & \sum_{l_{1}^{\;},l_{1}^{{}^{\prime }},l_{1}^{{}^{\prime \prime }},l_{1}^{{}^{\prime \prime \prime }}}\sum_{\mu^{\;},\mu^{{}^{\prime }},\mu^{{}^{\prime \prime }},\mu^{{}^{\prime \prime \prime }}}\int_{-\infty }^{\infty }\frac{d\omega^{\prime }}{2\pi }dV_{\nu \mu \mu^{\prime }}^{\left(3\right)l_{2}^{\;}l_{1}^{\;}l_{1}^{{}^{\prime }}}D_{\mu \mu^{\prime \prime }}^{≷,\;l_{1}^{\;}l_{1}^{{}^{\prime \prime }}}\left(\omega +\omega^{\prime }\right)\\ \times & D_{\mu^{\prime \prime \prime }\mu^{\prime }}^{≶,\;l_{1}^{{}^{\prime \prime \prime }}l_{1}^{{}^{\prime }}}\left(\omega^{\prime }\right)dV_{\mu^{\prime \prime }\mu^{\prime \prime \prime }\sigma }^{\left(3\right)l_{1}^{{}^{\prime \prime }}l_{1}^{{}^{\prime \prime \prime }}l_{3}^{\;}}, \end{array} \end{array}$$
with
(32)$$\begin{array}{c} \begin{array}{c} dV_{\nu \mu \mu^{\prime }}^{\left(3\right)l_{2}^{\;}l_{1}^{\;}l_{1}^{{}^{\prime }}}=\frac{\partial^{3}}{\partial R_{\nu }^{l_{2}^{\;}}\partial R_{\mu }^{l_{1}^{\;}}\partial R_{\mu^{\prime }}^{l_{1}^{{}^{\prime }}}}V_{anh}, \end{array} \end{array}$$
where the indices μ,ν, and σ designate atomic positions and $dV_{\nu \mu \mu^{\prime }}^{\left(3\right)l_{2}^{\;}l_{1}^{\;}l_{1}^{{}^{\prime }}}$ is a simplified notation for the third derivative of the VFF anharmonic potential energy $V_{anh}$ with respect to the νth atomic position in the $l_{2}^{\;}$ direction (*x*, *y*, or *z*) to the μth atomic position in the $l_{1}^{\;}$ direction and to the $\mu^{\prime \;th}$ atomic position in the $l_{1}^{{}^{\prime }}$ direction. This term couples together the greater/lesser Green’s functions $D^{≷}\left(\omega +\omega^{\prime }\right)$ and $D^{≷}\left(\omega^{\prime }\right)$, associated to the high-energy and low-energy phonons, respectively.

Like in the previous case, we restrict ourselves to the diagonal approximation with the anharmonic-force parameters calibrated to experimental data [51] and neglect the real part of the scattering self-energy since it mainly contributes to an energy renormalization [39,40,60]. In order to calculate the third-order expectation values of the thermal currents, three rescaling factors are injected into Equation (Equation 31). Without temperature difference between the left and right ends of the NW, the structure of interest is in equilibrium at room temperature. We then apply a small temperature difference (ΔT = 0.1 K) between both extremities of the NW to make the phonon thermal current flow.

First, we calculate the ballistic thermal current flowing through the NW structure in which the anharmonic phonon–phonon scattering is “turned off”. Then, to include the anharmonic effects, we compute the thermal currents through the conventional SCBA scheme, LOA + Padé, and LOA + Padé + Richardson. Figure 7 shows a comparison between the three methods. It is shown that all the LOA + Padé currents satisfy the current conservation law since the non-conserving diagrams are removed by the rescaling technique. However, the LOA currents diverge far from the SCBA value (not shown), testifying the importance of the anharmonic scattering even at room temperature. The application of Padé 0/1, 1/1, and 1/2 approximants results in the convergent behavior of the results, with 65%, 80%, and 87% accuracy with respect to SCBA, respectively. By applying the first-order Richardson extrapolation to the LOA + Pade 0/1 and 1/2 values, we improve the result, obtaining an accuracy higher than 90% without any loss in the numerical efficiency, compared to the SCBA result. Particularly, the LOA + Padé 1/2 + Richardson approximation requires 6 SCBA iterations, while the full SCBA loop requires 13 iterations to achieve the convergence. More exhaustive investigations of the calculation of anharmonic thermal currents in circular Si and GeNWs at several temperatures are reported in Reference [34].

## 4. Conclusions

In this work, we reviewed the theory, the so-called LOA approach combined with Padé approximants, developed within the NEGF formalism for the treatment of inelastic scattering. We benchmarked the method against the conventional SCBA approach by using an atomistic quantum transport code.

Differently from the SCBA, the LOA only includes current-conserving Feynman diagrams. This can results in a faster approximation of a current-conserving self-energy fulfilling the Dyson equation. We have demonstrated this case by investigating electron–phonon and anharmonic phonon–phonon scattering in a square Si GAA NW-FET. In the calculation of electronic and thermal currents, the LOA + Padé or the LOA + Padé + Richardson, implemented by a simple rescaling technique, successfully reproduced the SCBA values with a high accuracy (>90%) and computational efficiency (×6 faster for electronic currents and ×2 for thermal currents).

The main drawback of the approach is that the scaling factors used to calculate the higher-order perturbation terms (for both electron–phonon and phonon–phonon interactions) are determined via a rather pragmatic approach. Further work might be required to improve the choice of those scaling factors based on a more physically sound method.

Moreover, the method can be particularly advantageous (and thus worth to be investigated) in systems including localized states weakly coupled to the contacts, as for a deep quantum well. In these cases, the SCBA can take several thousands of iterations to converge and the use of LOA + Padé might be even more relevant.

Finally, it is worth noting that the reviewed method is also applicable to computationally more expensive codes such as a DFT-NEGF quantum transport simulator. In particular, it is expected that the LOA + Padé approach can provide more efficient results for device configurations where complex materials, e.g., transition-metal dichalcogenides, are involved, since the materials need an ab initio level of accuracy to describe their electronics properties and is, thus, costly. The approach should then faithfully broaden the accessibility of atomistic quantum transport codes based on the NEGF framework to investigate 3D realistic systems without the need for heavy computational resources.

## Figures and Tables

**Figure 1 materials-13-00060-f001:**
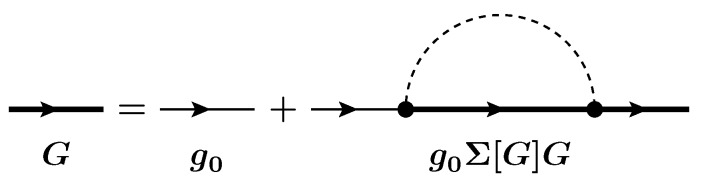
Feynman diagrams for the Dyson equation showing the relation between the noninteracting electron Green’s function $g_{0}$ (thin lines with an arrow) and fully interacting one *G* (thick lines with an arrow): The dashed line denotes the free phonon propagator.

**Figure 2 materials-13-00060-f002:**
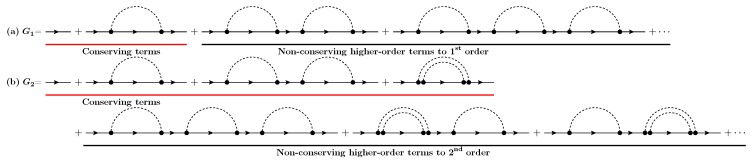
Feynman diagrams for self-consistent Born approximation (SCBA) Green’s functions, (**a**) $G_{1}$ (at first iteration) and (**b**) $G_{2}$ (at second iteration): Conserving (red-underlined) and non-conserving (black-underlined) terms are arranged according to ascending interaction order. Noninteracting electron Green’s functions are described by thin lines with an arrow while free phonon propagators are represented by dashed lines.

**Figure 3 materials-13-00060-f003:**
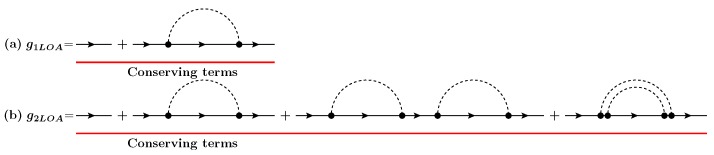
Feynman diagrams for Lowest Order Approximation (LOA) Green’s functions: (**a**) $g_{1LOA}$ at the first order and (**b**) $g_{2LOA}$ at the second order in interactions. Conserving terms are red-underlined. Noninteracting electron Green’s functions are described by thin lines with an arrow while free phonon propagators are represented by dashed lines.

**Figure 4 materials-13-00060-f004:**
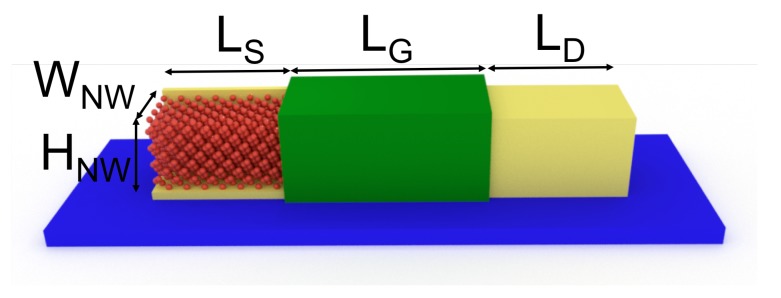
Schematic view of a Si gate-all-around (GAA) nanowire (NW) field-effect transistor (FET) crystallographically oriented along the 〈100〉 direction with a $H_{NW}\left(3\;\mathrm{nm}\right)\times W_{NW}\left(3\;\mathrm{nm}\right)$ square cross section: (Electron transport) The gate length is $L_{G}=13$ nm, and the length of source/drain region measures $L_{S/D}=10$ nm. The NW structure is surrounded by a 1-nm-thick silicon dioxide layer. The concentration of donors in the source and drain regions is $N_{S/D}=1\times 10^{20}\;\;{\mathrm{cm}}^{-3}$. (Phonon transport) The NW total length ($L_{G}+L_{S}+L_{D}$) is 60 nm. The NW is undoped, ungated, and free of the oxide layer.

**Figure 5 materials-13-00060-f005:**
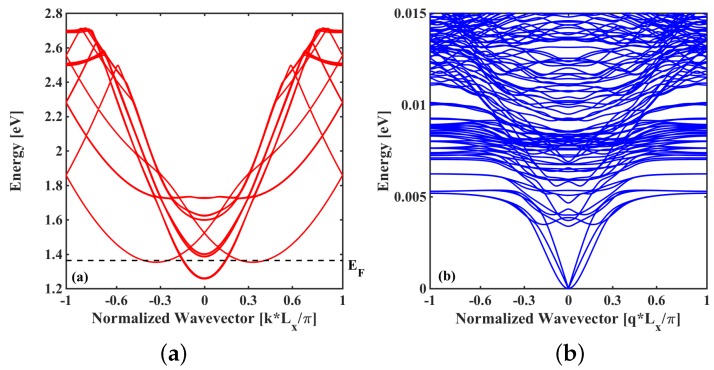
(**a**) Electronic conduction band-structure of the Si NW sketched in Figure 4, obtained with a full band tight-binding *sp*${}^{3}$*d*${}^{5}$*s*${}^{*}$ model without spin-orbit coupling. (**b**) Phonon dispersion relation obtained with a modified valence-force-field method. $L_{x}$ is a slab length.

**Figure 6 materials-13-00060-f006:**
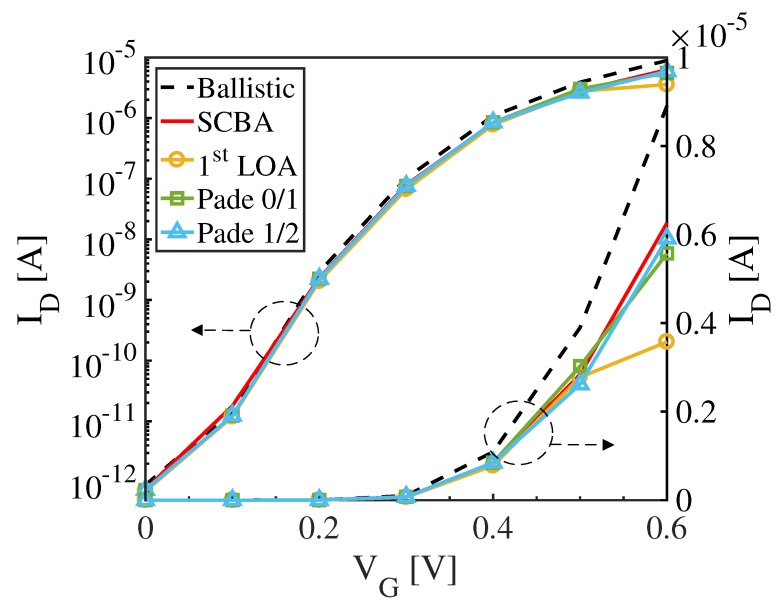
$I_{D}-V_{G}$ transfer characteristics of the n-type 3 nm × 3 nm square cross-sectional Si GAA NW-FET obtained for ballistic regime and by SCBA, 1st-order LOA, Padé 0/1, and Padé 1/2.

**Figure 7 materials-13-00060-f007:**
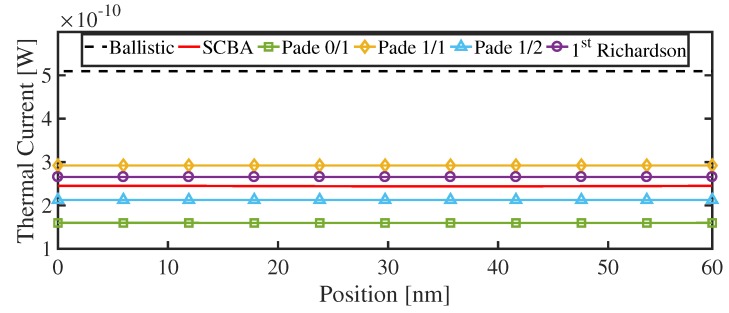
Thermal currents at room temperature in the 3 nm × 3 nm square cross-sectional Si NW of Figure 4 obtained for the ballistic regime and within the SCBA, Padé 0/1, Padé 1/1, Padé 1/2, and the first-order Richardson extrapolation.

**Table 1 materials-13-00060-t001:** Accuracy ($\varepsilon =100\times |I_{SCBA}-I|/I_{SCBA}$ where I is ballistic, LOA, Padé, or Richardson current) and efficiency (# of iterations) comparisons of 1st and 3rd LOA, Padé 0/1 and 1/2, and the corresponding Richardson currents with ballistic and SCBA currents at $V_{G}=0.6$
V.

	Ballistic	LOA1	LOA3	Padé 0/1	Padé 1/2	Richardson	SCBA
Current [A]	8.9 × 10${}^{-6}$	3.57 × 10${}^{-6}$	−2.616	5.6 × 10${}^{-6}$	5.9 × 10${}^{-6}$	6.1 × 10${}^{-6}$	6.3 × 10${}^{-6}$
ε [%]	42.1	42.0	4.2e7	10.8	6.4	2.0	0.0
Number of iterations	0	1	6	1	6	6	35

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
