# Peer review of "Quantum Treatment of Inelastic Interactions for the Modeling of Nanowire Field-Effect Transistors"

_materials, 2019, doi:10.3390/ma13010060_

Round 1

Reviewer 1 Report

The authors reviewed LOA approach combined with Pade approximants for the treatment of inelastic scattering. The performance of reviewed method were benchmarked with conventional SCBA approach on Si GAA NW-FET. Results show the efficient and effective calculation for inelastic scattering in nanostructures. The topic is interesting and its potential application may also be extending to more complex materials and device systems by reliving the computational burden by conventional method.

Overall, this manuscript is prepared well and could be considered for publication if authors can also discuss in brief about the shortcomings or potential future improvement on the reviewed method. 

Author Response

In the following, our answers are integrated in the original comments:

The authors reviewed LOA approach combined with Pade approximants for the treatment of inelastic scattering. The performance of reviewed method were benchmarked with conventional SCBA approach on Si GAA NW-FET. Results show the efficient and effective calculation for inelastic scattering in nanostructures. The topic is interesting and its potential application may also be extending to more complex materials and device systems by reliving the computational burden by conventional method.

Reply:

We would like to thank the Reviewer for his/her positive and fruitful comments.

Q1:

Overall, this manuscript is prepared well and could be considered for publication if authors can also discuss in brief about the shortcomings or potential future improvement on the reviewed method. 

Reply:

We added the following paragraph in the conclusion of the paper to describe the potential future improvement and opportunity of the reviewed method:

“The main drawback of the approach is that the scaling factors used to calculate the higher order perturbation terms (for both electron-phonon and phonon-phonon interactions) are determined via a rather pragmatic approach.  Further work might be required to improve the choice of those scaling factors based on a more physically sound method.

Moreover, the method can be particularly advantageous (and thus worth to be investigated) in systems including localized states weakly coupled to the contacts, as for a deep quantum well. In these cases, the SCBA can take several thousands of iterations to converge and the use of LOA-Padé might be even more relevant.”

Reviewer 2 Report

The authors review their decade-long program to optimize the computation of the effects of electron-phonon scattering in nano-wire field effect transistors (NWFET's). The crux of their program is to select out "conserving" Feyman diagrams from those generated by the self-consistent Born approximation (SCBA), and to thereby improve computational efficiency.  They succeed in reproducing results from SCBA, which require up to 35 interactions, with their conserving-Feynman-diagram approach with only a handful of iterations.  These results are clearly presented in the manuscript, which I believe merits publication.

The author's may consider answering the following questions in a revised manuscript, however.

1.The SCBA, as well as the optimized conserving-Feynman-diagram approach followed by the authors, notably does not include "crossing" two-phonon processes.  In other words, the authors neglect vertex corrections.  What precisely is the small parameter in their model for the electron-phonon interaction in NWFET's, Fig. 5 and Eqs. 28-30, that permits them to invoke Migdal's theorem?  Is it again the square root of the ratio of the electron mass to the ionic mass, or do the 1D quantum-confinement effects of the nanowire change that result?  In particular, can vertex corrections be neglected in the "strong electron-phonon scattering" regime at 0.6 V of gate voltage for the NWFET that is displayed by Fig. 6?

2. Why is the valence band structure not displayed by Fig. 5(a)? Do holes make no contribution to the electric currents? Is the Fermi level at zero energy?  And "k [normalized]" and "q [normalized]" are not defined in the caption to Fig. 5.

3. And I found the following typos.

line 24 - remove comma after "engineering"

line 176 - replace "elctronic" with "electronic"

line 197 - replace "expecttion" with "expectation"

line 216 - replace "able" with "Table"

Author Response

In the following, our answers are integrated in the original comments:

The authors review their decade-long program to optimize the computation of the effects of electron-phonon scattering in nano-wire field effect transistors (NWFET's). The crux of their program is to select out "conserving" Feyman diagrams from those generated by the self-consistent Born approximation (SCBA), and to thereby improve computational efficiency.  They succeed in reproducing results from SCBA, which require up to 35 interactions, with their conserving-Feynman-diagram approach with only a handful of iterations.  These results are clearly presented in the manuscript, which I believe merits publication.

Reply:

We would like to thank the Reviewer for his/her positive and fruitful comments.

The author's may consider answering the following questions in a revised manuscript, however.

Q1:

1.The SCBA, as well as the optimized conserving-Feynman-diagram approach followed by the authors, notably does not include "crossing" two-phonon processes.  In other words, the authors neglect vertex corrections.  What precisely is the small parameter in their model for the electron-phonon interaction in NWFET's, Fig. 5 and Eqs. 28-30, that permits them to invoke Migdal's theorem?  Is it again the square root of the ratio of the electron mass to the ionic mass, or do the 1D quantum-confinement effects of the nanowire change that result? In particular, can vertex corrections be neglected in the "strong electron-phonon scattering" regime at 0.6 V of gate voltage for the NWFET that is displayed by Fig. 6?

Reply:

We thank the reviewer for raising this very important point.

Strictly speaking, the SCBA and the LOA-Padé approach can be exact only for the description of one-phonon processes since, although they include certain multi-phonon processes they ignores those coming the vertex diagrams. Therefore, those approaches are rigorously valid as long as only one-phonon processes are involved. Nevertheless, in this work, assuming that vertex corrections are negligible, we investigated the validity of the LOA-Padé approach in realistic nanowire transistors, using the SCBA as a benchmark.

We can also note that even at 0.6 V of gate voltage for the NWFET that is displayed by Fig.6, the “electron-phonon” is not that important since it induces a current decrease of less than 30 %.

Finally, we agree with the reviewer that in some specific cases the vertex corrections are not negligible anymore. However, the validity of Migdal's theorem is always assumed in the simulation of electronic devices, and checking whether this is actually justified is beyond the scope of the paper. 

Q2:

Why is the valence band structure not displayed by Fig. 5(a)? Do holes make no contribution to the electric currents? Is the Fermi level at zero energy?  And "k [normalized]" and "q [normalized]" are not defined in the caption to Fig. 5.

Reply:

In this work, only n-type nanowires and then electron transport through the conduction band are considered. This is why we do not display the valence band structure. Study about the hole transport in the valence band has been also reported. Please see ref.[33] of the present manuscript: Lee, Y.; Bescond, M.; Cavassilas, N.; Logoteta, D.; Raymond, L.; Lannoo, M.; Luisier, M. Quantum treatment of phonon scattering for modeling of three-dimensional atomistic transport. Phys. Rev. B 2017, 95, 201412(R). The Fermi level is now indicated on Fig. 5 as well as the normalization of the wave vectors.

Q3:

And I found the following typos.

line 24 - remove comma after "engineering"

line 176 - replace "elctronic" with "electronic"

line 197 - replace "expecttion" with "expectation"

line 216 - replace "able" with "Table"

Reply:

We thank the reviewer. The typos have been fixed.

Reviewer 3 Report

Aside from some minor edits for English and proofreading, this is a well-written review of the Lowest Order Approximation for quantum simulations. The descriptions of the methods are complete and clear, and previous work is well referenced.

Author Response

In the following, our answers are integrated in the original comments:

Aside from some minor edits for English and proofreading, this is a well-written review of the Lowest Order Approximation for quantum simulations. The descriptions of the methods are complete and clear, and previous work is well referenced.

Reply:

We would like to thank very much the Reviewer for his/her positive and fruitful comments.

The typos have been fixed.